# The Actualization of the Transplantation Complex on the Axis of Psychosomatic Totality—Results of a Qualitative Study

**DOI:** 10.3390/healthcare9040455

**Published:** 2021-04-12

**Authors:** Marie Eichenlaub, Barbara Ruettner, Annina Seiler, Josef Jenewein, Annette Boehler, Christian Benden, Uwe Wutzler, Lutz Goetzmann

**Affiliations:** 1Department of Psychology, Medical School Hamburg MSH, 20457 Hamburg, Germany; barbara.ruettner@medicalschool-hamburg.de; 2Department of Radiation Oncology and Competence Center for Palliative Care and Department of Consultation-Liaison Psychiatry and Psychosomatic Medicine, University Hospital Zurich and University of Zurich, 8091 Zurich, Switzerland; annina.seiler@usz.ch; 3Department of Medical Psychology and Psychotherapy, University Hospital of Graz, 8036 Graz, Austria; josef.jenewein@medunigraz.at; 4University of Zurich, 8006 Zurich, Switzerland; annette.boehler@bluewin.ch; 5Faculty of Medicine, University of Zurich, 8006 Zurich, Switzerland; christian_benden@yahoo.de; 6Clinic for Psychosomatic Medicine and Psychotherapy, Asklepios Fachklinikum Stadtroda, 07646 Stadtroda, Germany; u.wutzler@asklepios.com; 7Institute of Philosophy, Psychoanalysis and Cultural Studies (IPPK), 12047 Berlin, Germany; goetzmann@ippk.de

**Keywords:** lung transplantation, unconscious processing, transplantation complex, axis of psychosomatic totality, imaginary zone, grounded theory, depth-hermeneutic method

## Abstract

Although transplantation medicine is not new, there is a clinically justified gap in the existing literature with respect to the psychological processing of lung transplants. The present study aims to examine whether lung transplantation leads to an actualization of psychological, e.g., oral-sadistic fantasies. Following a qualitative approach, 38 lung transplant patients were interviewed three times within the first six months after transplantation. Data analysis focused on identifying unconscious and conscious material. The inter-rater reliability for all codes was calculated using Krippendorff’s Alpha (c-α-binary = 0.94). Direct and implicit evidence of a so-called *transplantation complex* was detected e.g., regarding the “incorporation” of the dead donor and his lungs. These processes occur predominantly at an imaginary level and are related to the body. Our findings emphasize that such psychological aspects should be borne in mind in the psychological treatment of lung-transplant patients in order to improve the processing of lung transplants, and that this might have a positive effect on patient adherence.

## 1. Introduction

A lung transplant represents a life-saving treatment option for terminal lung disease. Studies have shown that quality of life and well-being increase significantly after lung transplantation (lung transplant studies) [1,2,3,4,5,6,7,8,9,10,11,12]. Despite this, it is relatively unusual for certain aspects of the unconscious processing of a lung transplant to be addressed. One reason for this must surely be the challenge of the scientific operationalization of unconscious material. The first psychodynamic studies now date back 50 years [13,14,15,16]. Muslin [15,16] described the procedural internalization of the transplant, which has been confirmed by more recent studies [17]. An incomplete passage in terms of poor organ integration may result in significant psychological consequences (e.g., low adherence to immunosuppressive medication, with the risk of organ rejection) [18]. These studies also highlighted the special nature of the complex recipient-donor relationship, which may be influenced by the recipient’s early relationship experiences [19]. 

As part of the present research project, the unconscious processing of lung transplantation was recently investigated in a qualitative single-case study [20]. In this study, the postoperative dream and waking narratives of a lung-transplant patient were analyzed using the Zurich Dream Process Coding System [21,22]. Based on the different sources (waking narratives and dream reports), an unconscious *transplantation complex* was reconstructed. Unconscious complexes are stored in long-term memory and can be activated, for example, by transplantation [20,23]. They may include, among others, archaic oral sadistic fantasies. Such early and unconscious fantasies are to be understood as “structuring principles of psychic life” that determine each human individual ([24] (p. 903) [24,25,26]). Our fundamental assumption is that these unconscious fantasies organize all unconscious and conscious mental life. They are triggered and actualized by so-called “current concerns”, e.g., by the transplantation experience. Thus, both direct and indirect references to these fantasies can be found in the individual’s thoughts. Indirect references are produced by psychological defense mechanisms such as “displacement”, “condensation” or “negation”.

In this single-case study, the patient initially grappled with the donor’s death in waking narratives two weeks after the transplant operation. Possible reasons given by the patient for the donor’s death were a traffic accident or even a “massacre”. Further analysis of the interviews revealed that the “massacre” motif in the waking narratives also appeared in a dream dreamt by the patient immediately after the operation. In this dream, possibly as a reference to an act of killing, bushes were cleared with chainsaws, and wood was chopped (The donor is killed). The fact that, in the dream, the patient did not know who was chopping the wood and felt uninvolved himself was interpreted as a psychological defense. The chainsaws occurring in the dream possibly indicated that the patient’s ribs were being sawn open (The body is broken open). The patient then addressed a statement that transplants have something to do with cannibalism (Objects penetrate into the body/are devoured (incorporation)). Through this association, the possible reactivation of oral-sadistic fantasies [25,26] through the transplant operation became particularly clear, i.e., very early, unconscious fantasies were activated of a human being, including his organs, being eaten and living on in the other, in this case the inner world of the donor (The donor is part of the recipient’s internal world). In the third interview, held six months after the transplant operation, the fantasy that something foreign was part of the patient’s inner world became evident: The patient drew the comparison with an adopted child who is left in the dark as to the identity of his biological parents. The comparison with a pregnancy also occurred to the patient: In his first dream after the operation, the patient dreamt that he had moved into a new yet somehow familiar residence with his wife, just as the donor moved into the patient’s body with his lung (The donor is the recipient; identification, the (new) object is a member of the family of organs). Here too, the metaphor of pregnancy was used, which ultimately led, however, to a separation from the “baby” (The new organ (lung) can be expelled). The will to survive and the incorporation of the lung can thus be associated with fears of (organ) rejection [20]. Based on this qualitative data analysis, the dream and waking narratives were used to reconstruct a so-called psychological *transplantation complex*, composed of unconscious or preconscious fantasies around the transplant experience, and including the following aspects: -The donor is killed,-The body is broken open,-Objects penetrate into the body/are devoured (incorporation),-The donor is the recipient (identification),-The donor is part of the recipient’s internal world,-The (new) object is a member of the family of organs,-The new organ (lung) can be expelled.

In the present study—a single-case analysis—the question arises as to how typical the *transplantation complex* and the corresponding activation of unconscious fantasies (identified in an individual patient and probably stemming from early life) might be of the transplant patients in the overall sample of the research project (*N* = 38). The extent to which the *transplantation complex* is directly or at least indirectly represented in the narratives is examined. In addition, we attempt to determine the semantic level at which the *transplantation complex* is represented, i.e., whether the physical experience is symbolic, asymbolic, or imaginary (Axis of Psychosomatic Totality) [27]. By symbolic physical experience we mean that a physical symptom has an unconscious meaning (bodily experience at the symbolic pole). In the case of the asymbolic experience, the physical symptom has no symbolic meaning (bodily experience at the asymbolic pole). Physical illnesses, such as heart, circulatory and lung diseases are found at the organic pole. An imaginary structured experience directly reflects the individual’s body image (bodily experience in the imaginary zone) [27]. 

## 2. Materials and Methods

### 2.1. Sample and Data

Recruitment for this prospective longitudinal study was conducted at the University Hospital Zurich (USZ) from 2012 to 2014. The study was approved by the Ethics Committee of the Canton of Zurich. Patients were informed about the study and subsequently signed a statement of agreement. The inclusion criterion was having undergone a lung transplant two weeks previously. The exclusion criteria were being under 18 years of age, having insufficient knowledge of German, and being in an unstable condition of physical health (e.g., suffering from dyspnea, inflammation, pain or fatigue) which would preclude participating in an interview and completing questionnaires. One of the patients had to be excluded owing to a critical health condition and repeated admission to the intensive care unit (ICU). Furthermore, we checked in advance whether delirium, dementia or psychosis had been diagnosed. All other patients with psychological distress were included, since the data of all lung-transplant patients was considered relevant.

The longitudinal sample consisted of 47 lung-transplant patients. Because of the exclusion criteria, the effective sample of the present study comprised 38 patients. Participants included 20 male and 18 female lung-transplant recipients with a median age of 47.67 years (range 20–68 years) at the time of the transplant operation. In our clinical sample, the main indication for transplantation was chronic obstructive pulmonary disease (COPD; 39.47%), followed by cystic fibrosis (34.21%). All 38 patients were still alive six months after transplantation. Detailed patient demographics and clinical characteristics are summarized in Table 1.

Semi-structured interviews were conducted two weeks (t_1_), three months (t_2_) and six months (t_3_) after lung transplantation by two USZ psychologists. The interviews, which averaged 15 min in length, were audio-documented, transcribed according to the transcription rules of Kruse [28] and Selting [29], and imported into the software program atlas.ti (version 8.4.5, Mac). Questions were on the topics of transplantation, lung, donor and medication. A detailed description of the key questions can be found in Seiler and colleagues [10].

### 2.2. Data Analysis

The interviews were analyzed for content against the backdrop of grounded theory [30] and for unconscious processes using depth hermeneutics [31,32]. “Grounded theory” describes a method that “classifies units of meaning within an encompassing horizon of meaning” [33] (p. 44). In this study, the interviews were analyzed using theory-led deductive structural codes. This means that codes were deduced from the concept of the *transplantation complex* and the theory of the axis of psychosomatic totality (regarding the semantic, i.e., symbolic, asymbolic or imaginary nature of physical experience). These codes were defined in a theory-led manner and used to code the patients’ statements. Four interviews were then coded and a codebook with definitions and anchor examples created. 

Because the case study was predominantly dream material but most of the other patients reported little or no dreaming, we decided to examine the narratives in terms of both direct and indirect cues relating to the *transplantation complex*. Direct cues are immediate namings of some aspect of the *transplantation complex*, such as when a patient would state that the donor continues to live on within him, or that he has taken on traits of the donor. The fantasy of being to blame for the donor’s death is also a direct reference to the *transplantation complex*. Thus, in the case of direct references, it is assumed that the patient is aware of what is being said and that the corresponding fantasies are “subject to the reality principle” [34] (pp. 607–608) [35] (p. 398). Such fantasies, however, are often not admitted because they seem alarming or even morally unacceptable. In the utterances, then, only indirect, implicit references are made to the fantasies, which are usually repressed. Indirect references can be found e.g., in dreams, in which unconscious fantasies are subjected to a so-called primary process. These unconscious fantasies are processed by defensive mental operations such as “displacement”, “condensation” or “negation” [34] (pp. 607–608) [35]. In the defense mechanism of “displacement”, aggressive or libidinous impulses may be shifted from one person to another. In “condensation”, several representations merge into new imaginative content [34,35]. “Negation” can also be used, i.e., conflict-laden material becomes conscious only in the negated form [20,36,37]. The waking narrative can then “provide clues to the depth of meaning of a dream” [20] (p. 519). 

In the present study, the coefficient of agreement (inter-rater reliability) in terms of both the direct and indirect references to the *transplantation complex* and the semantic level of bodily experience was calculated with Krippendorff’s alpha coefficient (c-α-binary). A strong agreement of c-α-binary 0.93 (min = 0.85, max = 1.0) [38] among raters (relative to 20% of the randomly selected quotations for each code) can be seen. 

Table 2 shows the deductive structural codes, code definition and inter-rater reliability.

## 3. Results

### 3.1. Results: Evidence for the Transplantation Complex

Both direct (54.83%) and indirect (45.17%) references to the *transplantation complex* are found. An exception is the aspect that the body is broken open, which is only represented in an indirect manner. Incorporation (29.51%) as well as rejection of the organ (22.22%) are presented most frequently. 

The donor is killed: The fantasy that the recipient was killed occurs in 14.49% of cases. This is a transplant-specific conflict: It is only through someone’s death that the recipient continues to live. The death wish is presented directly and indirectly, partly in a negated form, in which the patient describes that he is not to blame for the death of the donor, but at the same time benefits from his death: “I can’t help it that he died. I’m benefiting from someone who died. I imagined that he might be buried” (Mr. A, t_1_). Indirectly, this aspect is presented when patients talk about the topic of death in general, e.g., Mrs. B (t_1_), who “[found] the contrasts between life and death crazy.”. The theme was sporadically presented through dream reports: “An evil woman came up to me and suddenly turned into a doll-like figure and wanted to kill me” (Mrs. B, t_1_). In this context, the doll-like figure as manifest content could represent the organ donor. 

The body is broken open: Patients rarely speak of the body being broken open, and only then in a direct manner (0.55%). One patient, for example, reports that he “dreamt of power saws”. These stood for the fact that in a transplant, saws would be used and the ribs would have to be broken (Mr. P, t_1_).

Objects penetrate into the body/are devoured (incorporation): This aspect is mentioned by the highest percentage of patients (29.51%), and is mainly presented indirectly. The patients generally state that objects are incorporated. The majority of them also report on the intake of medication, which could represent manifest content for the incorporation of the new organ. Medication intake is overwhelmingly associated with negative aspects: “It’s an evil because I have to swallow so many pills. That’s why it’s is a necessary evil for me” (Mr. C, t_1_). The aspect of objects being eaten is found again in one patient, in the form of a dream. Mrs. D describes how worms wanted to devour her. The worms could represent a manifest cue for the donor who wants to recover his stolen organ. In addition, she reports that the mattress on which she is lying constantly fills up with air, then deflates. This account can be seen as indicative of the latent notion of internally perceived brokenness. The filling up with air could represent incorporation; the emptying could indicate the aspect that the new organ (lung) can be expelled. 

“I had dreams of worms trying to eat me. I probably had these nightmares because of the medication. I was lying on a decubitus mattress that was constantly filling up with air and deflating. In the dream, I felt like worms were coming out of this mattress, wanting to eat me” (Mrs. D, t_2_). 

There are few direct references in which patients describe the lung being incorporated into the body, e.g., “the lung is now inside me” (Mrs. E, t_1_).

The donor is the recipient (identification): Patient identification with the deceased donor is apparent in only a few reports (3.83%). Direct references show that the representations of the self and the recipient, described here as a “part”, become one with the representations of the self: “And that’s it, now I’m a part of him and he’s a part of me” (Mr. F, t_2_). Identification is evident when patients report on adopted characteristics of the donor. One participant described needing to cry more since the transplant. His son commented that he had been given the lung of a crybaby (Mrs. G, t_2_). Other patients reported a feeling of kinship with the donor: “I feel like I’m related to him” (Mr. H, t_1_). Indirect cues represent identification or comparisons with other objects. Frequently, this involves a comparison of the old and new lungs, as in the case of Mr. I (t_1_), who reported that his “old lung was exchanged for a new lung.”. In this quote, a comparison of the recipient’s person or his old lung with another object (the donor’s lung) is evident. 

The donor is a part of the recipient’s internal world: The feeling that the donor is part of the inner world of the patient occurs in 13.3% of cases. Direct references are evident when the donor is situated directly in the inner world of the recipient. Thus, Mr. F (t_1_) describes how he has a part of the donor and the donor is also part of him. “I am connected with him. He lives on a little bit through me, as I do through him, you could say.”. Indirect references predominate, in which not the donor himself but his donated organ becomes a part of the recipient’s inner world: “The new lung belongs to me. It belongs to my body. It’s a part of me and I‘ve accepted it. It’s inside me, breathing” (Mrs. J, t_1_). 

The new object (lung) is a member of the family of organs: The idea of membership is presented for the most part directly and is the third most frequent aspect of the *transplantation complex* (15.66%). Some patients see the new lung as a member of the family of organs. One patient speaks of having adopted the new lung. She does not want to lose it as a new member of the organ family: 

“From the start, I adopted it as my own. It even has a name. And yes, I have it inside me from her (the donor). It’s already a part of me. I don’t experience it as a foreign body at all. I’ve adopted it lock, stock and barrel. It’s my lung now and I’m not giving it away” (Mrs. K, t_1_). 

The aspect is displaced when the whereabouts of objects in the body become clear. For example, when patients talk about constipation: “My digestive system doesn’t work well then, and I suffer from constipation” (Mrs. L, t_1_). The manifest content (constipation) could indicate that there is a desire for the object (new lung) to remain in the body. In terms of displacement, it could be assumed that the organ is or remains a member of a group. 

The new organ (lung) can be expelled: Reports that the new organ can be expelled are the second-most frequent (22.22%). Here, patients speak for the most part directly about the (fantasized) rejection of the organ. Mr. M (t_2_), for example, describes how he “would mull over whether rejection would or would not occur, whether he would pick up a bug or not”. He would try to be careful, he said, and yet he had no control over whether rejection would occur or not. The aspect is presented indirectly when objects are expelled or excreted from the body. Diarrhea (“I have continuous diarrhea”) (Mr. C, t_1_) as well as vomiting (“Yesterday I had to throw up, today I was able keep it in”) (Mrs. N, t_1_), are mentioned particularly often, and can be understood as indirect references to the general loss of an object, such as the lung. Another indirect reference is shown, for example, by a patient’s dream. The manifest content that something is being ripped out (venous catheter) could indicate the fear that the new lung is no longer a member of the organ family and will be rejected: 

“I had been having nightmares for a long time. I dreamt that my caregiver came into my room and ripped out my venous catheter. I then woke up and actually saw my caregiver in front of me and then immediately flinched and had to reassure myself that the CVC was still there” (Mrs. O, t_1_). 

### 3.2. Results of Co-Occurrence-Analysis: Transplantation Complex and Axis of Psychosomatic Totality

The *transplantation complex* is for the most part presented in the imaginary zone (98.4%) and only occasionally at the organic pole (1.6%). Aspects are not presented at the symbolic and asymbolic pole. When the *transplantation complex* is embedded in the imaginary zone, the physical experience of the patients may include physical and psychological aspects at the body-image level. Below, we show the results of the co-occurrence analysis, which relate to the individual aspects of the *transplantation complex*:

The body is broken open: An indication that the body is broken open can be linked to the imaginary zone (0.81%). The direct link with the physical becomes clear when Mr. A (t_3_) reports that the scar and his ribs hurt. The reason for this is presumably that his ribs had to be broken for the operation. The fact that this resonates with the psychological experience of violent incorporation, which alters body image, speaks for the coding of the imaginary zone.

Objects penetrate into the body/are devoured (incorporation): The incorporation of objects (lung) occurs most frequently in the imaginary zone (39.84%). Here, the mental image of something foreign penetrating into the body seems to be primarily associated with medical or physical processes such as the ingestion of medication. In the case of Mr. C’s quote (t_1_), the account of objects entering the body through medication intake is presented and includes the mental experience—survival is contingent on the incorporation of an object:

“It’s an evil because I have to swallow so many pills. But it’s an evil, period. That’s why it’s a necessary evil for me (...). Necessary so that I can continue to live at all. I’m attached to life. I have a wife, a nice house. We have a great time together. Why should I give that up? Because of these pills? No, I’d rather swallow these tablets (...). This is the focus of my life. Because I’ve got to take these things” (Mr.C, t_1_).

Another physical process (inhaling) also shows an associated increase in mental exertion, which is why the imaginary zone was coded:

“Meanwhile, the pill-swallowing is going very well. The only thing that still causes problems is inhaling. This fungizone always prickles my throat so hard, it’s just such an uncomfortable feeling that sometimes I have to skip an inhalation because I just can’t bring myself to do it” (Mrs. P, t_2_).

The incorporation of objects (lungs) also occurs at the organic pole (1.63%), although this linkage is very rare compared to the imaginary zone. The quote from Mr. C (t_3_) shows the linkage: “Inhaling sometimes triggers shortness of breath in me and it prickles my throat. It’s really disgusting. That’s why I inhale for five days and then take a break again for five days”. As an indirect indication of object intrusion, inhalation is accompanied by stress in the form of shortness of breath (organic pole).

The donor is a part of the recipient’s internal world: The fantasy that the donor is part of the patient’s inner world is linked to the imaginary zone in a few cases (16.26%). It becomes clear that the donor is not located in the recipient’s internal world as a person, but only through his body part (the donated organ). Neither the donor nor his organ is perceived as a foreign body: “Well, actually I don’t worry too much that I’m carrying a foreign body here, so to speak” (Mr. I, t_1_). The new lung is experienced much more as a body part in its own right: “The lung is inside me, it belongs to me now” (Mr. R, t_1_). Through the quotes, it is clear that the physical (donor organ) contains the psychological experience and alters the patient’s inner world.

The new object (lung) is a member of the family of organs: Experiencing the lung as a new member of the organ family is the third-most common occurrence in the imaginary zone (19.51%). The imaginary zone can be identified when the imaginary space contains images or ideas about the integration of the new organ (lung) into the organ family. In this case, the new organ is located directly in the patient’s body image and occupies a central place in the patient’s life: “The lung is a part of me, it belongs to me. It’s in my body—on account of that fact alone, the new lung occupies a central place in my life” (Mrs. J, t_3_).

The following quote emphasizes the importance of the new lung in the lives of lung-transplant patients. The patient’s inner world consists of the mental image of taking care of the new organ so that it is integrated into the organ family and not rejected. Here, the patient feels compelled to adapt daily life to the new organ and to take and inhale medications correctly:

Mrs. O (t_1_): “Yes, exactly. The lung is still my focus. But also, everything I do or don’t do, I do out of consideration or a sense of duty toward my lung. I inhale for the lung, I take medication because of the lung, there are also certain things I should refrain from doing since the transplant. I also do that for my lung. It’s just all about the new lung.”.

I: “Even today, 6 months after the transplant?”.

Mrs. O (t_1_): “Yes, of course. I think the new lung will always be my center, all my life I will have to see to it that my scores remain good, to prevent rejection.”.

In the case of the “member” thought, the link with bodily ideas (imaginary zone) is indirectly evident through the report of constipation or digestive complaints. This form of not wanting to let go or not wanting to surrender (constipation) is understood here as an indirect reference to the integration into the organ family. Indigestion can be seen as a physical experience that contains the psychological experience, such as the fear of losing the new object as a member of the organ family: “But my most sensitive point is digestion. It worries me when I am constipated and can’t go to the toilet properly, or I eat something wrong and then suffer from digestive upsets” (Mrs. O, t_1_).

The new organ (lung) can be expelled: Actual or fantasized organ rejection is linked to the imaginary zone in 23.58% of cases. The aspect is often represented indirectly by the physical symptomatology such as diarrhea, vomiting or phlegm, which can refer to the general loss of an object (lung). The inclusion of psychological details in this account of physical experience in the context of transplantation speaks for the identification of the imaginary zone. With Mrs. L (t_1_), the psychological experience refers to the experience of dependence on others. The perceived dependence seems to have arisen from the excretion of the object, displaced by the report of diarrhea: “Unfortunately it has now switched to the opposite, and now I almost have diarrhea and can’t hold it in because I’m not mobile yet and always have to ring the bell first and wait for someone to come and help me.”. In the case of another patient, it is clear that the fantasized expulsion (physical experience) is associated with great worry (as a psychological experience):

“My main concern is taking good care of my lung and not damaging it. I am very careful not to catch a virus—maybe I am a little overcautious. I disinfect my hands all the time. Yes, and in the beginning, I had quite a lot of phlegm, and I was very worried that something was wrong with the new lung” (Mr. Q, t_1_).

The donor is killed and the recipient is the donor (identification) are not linked either to the imaginary zone, or to other poles of the axis of psychosomatic totality.

## 4. Discussion

The aim of the present study was to provide a deeper understanding of how a lung transplant is processed psychologically. To this end, we investigated whether the findings of the single-case study such as the activation of early unconscious fantasies by the transplant [20] could be applied to the entire sample (*N* = 38). We also examined whether the *transplantation complex* is represented directly, or at least indirectly, in the patients’ narratives. Furthermore, we attempted to determine the semantic level on which the *transplantation complex* is represented.

In the interviews with the 38 patients, both direct and indirect references to the transplantation complex were found with the following aspects: The donor is killed, The body is broken open, Objects penetrate into the body/are devoured (incorporation), The donor is the recipient (identification), The donor is part of the recipient’s internal world, The (new) object is a member of the family of organs, The new organ (lung) can be expelled. We conclude from these findings that the activation of unconscious fantasies most likely originating in early life does indeed seem to be a typical psychological process in lung-transplant patients.

Below, we first discuss the various aspects of the *transplantation complex* that were identified in the interviews with the patients:

The donor is killed: A preoccupation with the death of the donor is not at all unusual in transplant patients, since they survive through the death of the donor, or rather, with the help of the deceased donor’s lung. For those on the waiting list for a lung transplant, the will for survival and the wish for another person to die soon and donate his or her organs are very closely related. Thus, the aspect the donor is killed occurs in many interviews in direct and indirect forms. The indirect presentation of this aspect may be due to the fact that, despite their desire to survive or their will to live, the patients do not actually want someone else to die “for them” or for their wish for the other person to die to come true. In the interviews, this conflict also leads to the frequent mention of feelings of guilt. This is a typical way for transplant patients to process their conflicting feelings concerning the donor’s death [10,17,39]. Here, feelings of guilt arise as a consequence of the fact that it is only through the death of the other person (which may be eagerly desired) that the continuation of one’s own life is made possible. The feeling of guilt seems to be so great or discomfiting in some patients that it is warded off by denial, e.g., by the patients stressing that they are not to blame for the donor’s death. Often, a condemning or punishing “super-ego” (in the sense of an inner ethical authority) is revealed in the interviews. One patient, for example, reported a dream in which the donor wanted to take back his organ and thus destroy the patient or punish her for the “theft”. The often fear-laden fantasy that the transplanted organ could be rejected might be understood by bearing in mind the feeling of guilt, in the sense, too, of “making amends” [24,25,26].

The body is broken open: Although the aspect termed the body that is broken open is mentioned in an invariably direct way, it is referred to comparatively rarely. This aspect possibly plays a subordinate role to the other aspects of the *transplantation complex*. The direct presentation can probably be attributed to the fact that the donor’s body is opened to remove the lungs and the recipient’s to implant them. Owing to the patient’s destructiveness toward the donor, however—as with the violation of body image by the operation—the “breaking open” of the bodies might be so traumatic that these fantasies exceed the patient’s ability to verbalize them, thus preventing the defense mechanisms of displacement or condensation from setting in. These notions of the body being broken open would be rooted in very early fantasies, discovered and described by Melanie Klein [25,26]. In general, unconscious fantasies form the central building blocks of psychic life and all interpersonal relationships. Furthermore, they also play a role in the development of psychopathological symptoms [24,40,41]. According to Melanie Klein [25,26], the child tries to get at the mother’s contents (milk) in an aggressive way, destructive in other circumstances, by e.g., biting her breast. These frightening fantasies regarding both one’s own body and that of the donor would tend to live on by splitting off outside of the patient’s experience so that direct references were rarely and indirect references never found.

Objects penetrate into the body/are devoured (incorporation): Most patients present the incorporation of the lung or donor in an indirect manner, e.g., they generally report difficulties with processes of incorporation, such as those related to taking medication or involving dietary restrictions. Occasionally, incorporation occurs in dreams. One patient described being eaten by worms in a dream. Then, she said, the decubitus mattress on which lay kept on filling with air, then deflating, like a leaky balloon. This can be understood as a reference to the latent notion of the inwardly perceived turmoil of incorporating a foreign object. The filling up could stand for the incorporation of the lung and the emptying for the repulsion of the new organ. Here, the strong ambivalence toward the incorporated object also becomes clear. This can be foreshadowed by the major challenge, widely described in the literature, of integrating foreign objects (lungs) that enter the body on both a physical and psychological level [13,17,19,42]. In any case, early archaic fantasies that might be actualized by the transplantation processes are echoed in the context of incorporation [24,25,26,41]. The idea of incorporating a foreign object originally belonging to a dead person (lung) can seem so uncanny that it can only be admitted indirectly (displaced), i.e., in connection with medication, food, or at most in dream images (worms, decubitus mattress). During our research interviews, the connection with the lung or the donor, which we evaluated as indirect, remains open to speculation. This connection could only be validated in the context of psychotherapy, i.e., in further in-depth conversations with the patients.

The donor is the recipient (identification): In the previous single-case study [20], the patient dreamt that he moved into a new but somehow familiar residence, just as the donor with his lungs moved into the patient’s body or mental world. Here, an identification with the donor was suspected. In the present study, overall, only a few mostly direct, but also, to a lesser extent, indirect indications of identification are found, e.g., where the recipient reports a feeling of kinship with the donor or the presumption that he or she has taken on the characteristics of the latter. Indirect references often reveal comparisons of the old and new lungs. In terms of displacement, comparisons of the recipient’s person (old lung) with another person or his or her object (new lung) are understood as indirect evidence of identification. The longitudinal section of the three survey time points showed that reports of identification with the donor decreased significantly over time. From this we might conclude that identification processes mainly occur in the first few months after a transplant operation, and that they come to an end as time passes. This is also consistent with the clinical observations of Muslin [15,16]: The patient increasingly perceives him- or herself as separate from the donor. With his concept of “appersonization”, Schilder [43] described a process in which a person identifies with the physical traits of the other, in our case with those of the donor: He speaks of “body intercourse” [43,44]. This “appersonization” could serve two purposes: Firstly, the modification of one’s own body image in order to restore the injured body schema [45]; secondly, an (unconscious) identification of the recipient with the donor, so that the recipient can experience his lungs as his own organ [17]: If the recipient is (like) the donor, the donor’s lungs also belong to the recipient.

The donor is a part of the recipient’s internal world: The idea that the donor is part of the patient’s inner world is for the most part presented indirectly, for example when patients report that the new lung now belongs to them, that it is “inside” them and that it breathes like an independent person. For the majority of patients, the donor is indeed located as part of the recipient’s inner world. With many patients, however, the donor is not mentioned by name but is included in the recipient’s inner world through the donated organ, e.g., when patients describe how the lung belongs to them and their body. Here too, the above-mentioned notion of identification can be assumed: The donor seems too threatening, and thus can only be absorbed into the recipient’s inner world with the help of defense mechanisms.

The new object (lung) is a member of the family of organs: The predominantly direct representation of the idea of membership is shown by patients being the proud owners of a new object and perceiving the new organ as a part of their own organ family, e.g., when they report that the organ has been well accepted and they no longer wish to give it up. Symbolization, e.g., giving the lung a name, which is seen in isolated cases, could help the patient to accept the lung into the organ family [17].

The patients’ main concern is that the lung could be rejected, i.e., it will cease to be a part of the organ family. The frequency with which transplant patients report constipation and digestive discomfort as side-effects of the variety of medications prescribed is striking [46]. From a psychological perspective, however, the frequent reports of constipation may not only have a physical significance, but may also refer to the psychological notion of the “membership thought” (in terms of displacement): In this context, constipation can be understood as a vehicle for addressing the retention of the lungs at the body-image level. Above all, the very frequent report of constipation can also be understood as a consequence of the fear that the new organ could be rejected [47,48].

The new organ (lung) can be expelled: The concept that the organ can be expelled is presented second-most often. This finding is not surprising. One of the greatest worries of lung-transplant patients is the possibility of losing the new organ through acute [49,50] or chronic [51,52] rejection. Recent analyses of both lung and heart transplants by the German Organ Transplant Foundation [53] suggest an 84.07% chance of survival one year after the operation, given known status. Two- and three-year survival odds are 79.84% and 72.51%, respectively [54,55], while the odds of surviving for five years after the transplant operation are 59% [56]. The most common cause of death (30 days postoperative) is primary graft dysfunction [57,58,59,60,61]. Because patients are aware that a transplant is chiefly a life-extending procedure, they often present their concerns about rejection in a direct manner.

The concept is presented indirectly when speaking about diarrhea or vomiting, which could generally represent the loss of the new organ. Through these symptoms, worries and fears that could implicitly refer to the lung become evident. Fear of rejection can be exacerbated by the fact that vomiting or diarrhea can impede the adequate absorption of immunosuppressive medications that prevent rejection.

Worry that the organ may be rejected is also occasionally evident in dream reports. Possibly, some patients refuse to engage with worry about loss of the organ because death seems too threatening. In this case, the theme of rejection is more likely to appear in the dream experience, where control over psychological defenses is usually less strong. One patient, for example, reports that in her dream, her caregiver ripped out her central venous catheter. In this context, the tearing out of the venous catheter could stand for a manifest indication of the general loss of an object such as the lung. The person who tears out the venous catheter in the patient’s nightmare is experienced as attacking her own body and robbing the recipient of her precious, vital contents. Here we see the archaic and existential fears of transplantation, reinforced by the actualization of early fantasies, with which transplant patients must often come to grips.

The co-occurrence analysis of the *transplantation complex* and the axis of psychosomatic totality showed that the *transplantation complex* is mainly presented in the imaginary zone, i.e., the *transplantation complex* is chiefly processed in an imaginary fashion. The French analyst Jacques Lacan held the view that the corporeal, i.e., the bodily image, presents itself mainly in the so-called imaginary psychological realm [62,63]. With reference to the present study, the body image is massively manipulated and violated e.g., by breaking open and transplanting the new organ [45]. The manipulative destruction of the body is also evident in Kleinian [25,26] theory, in which unconscious fantasies have a bodily basis and revolve thematically around bodily orifices. The importance of also considering the structure of the new body image during treatment is shown by the major role that body image also plays in mental health [63].

In particular through the breaking open of the body (the body is broken open), a significant change in body image takes place. This aspect is presented only very rarely, however, and is therefore only slightly embedded in the imaginary zone. We can therefore only presume that the idea of the body being broken open overwhelms the formative integrative power of the imaginary. Psychoanalytic literature describes how, in such a case, the imaginary of the body somehow disintegrates as if it were “slipping away” [62] (p. 129) [64] (p.118). For example, a patient reports the pain of the scar, which he attributes to the breaking open of his body. In this context, the physical pain could also refer to the pain and traumatic imagination of a violent opening of the body; it would be the pain of having (temporarily) lost bodily integrity [65].

Most frequently, however, the incorporation of objects is associated with the imaginary zone. This change in body image is perceived by many patients as a major psychological burden, especially due to the constant taking of medication, which is described as a “necessary evil.”. The idea that the body is altered by the foreign, stolen object of a dead donor, which could also reactivate fears of fusion with a threatening object, can also be burdensome [17,45,66]. On the other hand, the fact that integration processes are often modified at the bodily, so-called organic pole, reflects the fact that the experiences of transplant patients are actual interventions that take place at both the material, physical-bodily, and imaginary-bodily levels. This interaction is evident in one patient’s description of often experiencing shortness of breath and circulatory problems during inhalation (to care for the transplanted lung). These symptoms could be side-effects of the medication, but could also be triggered by anxiety in relation to unconscious fantasies of the manipulated body [25,26,46,47,48].

Evidence that the new organ is a member of the new organ family is very often linked to the imaginary zone. The fact that physical perception changes is indicated by a new organ becoming a member of the organ family, probably for the long term. One patient vividly describes how he must take care of the object for the rest of his life so that it is not rejected and remains a member of the organ family. The indirect references e.g., to constipation can be understood as a form of not wanting to let go or not wanting to give something up, or as a fear of having to give up the new organ and thus re-violate the imaginary body schema.

The second-most frequent (fantasized and feared) rejection reactions are those linked to the imaginary zone, since a rejection reaction would presumably alter the patient’s body image once more. The body image would also be manipulated if, owing to their physical limitations, patients experienced a sense of dependence on their caregivers. The fear here of a regression to early childhood developmental stages was noted in previous studies [13,45,67].

Neither the donor is killed nor the donor is the recipient (identification) is linked to the imaginary zone. This could be because these aspects are not responsible for the manipulation of the body image, and are thus not embedded in the imaginary zone. The death of or identification with the donor do not appear to be integrated into the recipient’s own body, and are thus not located as an external object in the bodily-imaginary world of the recipient.

### Study Limitations and Strengths

The small sample size (*N* = 38) is one of the limitations of this study. Moreover, since the sample was not examined to determine whether any mental disorders such as depression were present before the transplant operation, limitations in external validity should be mentioned. This was, however, the first time that such a large sample was qualitatively examined in terms of its unconscious processing in a longitudinal manner.

This unconscious processing could only be captured using manifest, direct and indirect cues, thus highlighting the limitations of operationalizing unconscious material not normally directly addressed by patients. Despite these methodological challenges, there is very good agreement of coding choices among the raters (inter-rater reliability).

## 5. Conclusions

The concept of a *transplantation complex* developed in a single-case study has been shown to be a sound approach to understanding how a lung transplant is psychologically processed. Although certain aspects will need to be re-evaluated in future studies, the results of this study have a number of implications for research and practice. Among others, they enabled the design of a questionnaire including the *transplantation complex* which should reduce the time needed for data collection and analysis. Because psychological vulnerability can be a predictor for poor organ integration [18], future studies should incorporate a comparison between patients with and without a psychiatric disorder (e.g., depression).

The results of the study—especially in terms of our understanding of unconscious processes in lung transplantation—are encouraging, and will be of benefit in developing new psychotherapeutic approaches for transplant patients. Further education and training, for example, could be geared to sensitizing the treatment team to the underlying problems of transplant patients: “What patients say represents only a conscious surface of the mental state” [68] (p. 1). To get below this surface, the treatment team should make space and time for transplant patients, and learn to “listen with a third ear” [69]. The indirect references to the *transplantation complex*, such as those revealed in dream reports, could be particularly helpful in this process. We therefore recommend talking to patients about their dreams and asking them to report them. The commonly occurring uncanny and strange-seeming fantasies (e.g., killing or incorporation fantasies) should be discussed to resolve feelings of guilt and fear. Here, the patient should be informed, as part of a psychoeducational approach, that these processes and fantasies are common, and are part of the process of successfully integrating the new organ. The newly acquired understanding could help to optimize patient quality of life and encourage adherence.

## Figures and Tables

**Table 1 healthcare-09-00455-t001:** Demographic and medical data of lung transplantation recipients.

Variables	LTx Recipients (*N* = 38) *n* [%] *^,a^
Male sex	20 [52.63]
Female sex	18 [47.37]
Third gender	none
Median age (range)	47.76 (20–68)
Single	14 [36.84]
Married	17 [44.74]
Divorced	6 [15.79]
Widowed	1 [2.63]
Indication for LTx	
COPD	15 [39.47]
Cystic fibrosis	13 [34.21]
Idiopathic pulmonary fibrosis	7 [18.42]
Other	3 [7.89]
Retransplant patients	2 [5.23]
Days in ICU, mean (range)	4 (2–29)
Weeks in hospital pLTx, mean (range)	4.5 (3–14)
Psychopharmacological intervention pLTx	10 [25]

* Percentages are rounded to the second decimal place. ^a^ Unless otherwise specified.

**Table 2 healthcare-09-00455-t002:** Codes, definition, anchor examples and Krippendorff’s alpha coefficients (c-α-binary)**.**

Codes	Definition	Anchor Examples	c-α-Binary *
Transplantation Complex	Direct and Indirect Cues		
The donor is killed	Direct cues: Reports that the donor has been killed. Indirect cues: The person who donates remains unnamed (subjectlessness or objectlessness); any form of killing; desires and fantasies of killing; typical conflicts relating to this theme.	“It’s a negative thing that the man had to die for the lung so that I could get the lung”; “I find the contrasts between life and death crazy.”	0.91
The body is broken open	Direct cues: Reports of the body being broken open by the transplant operation. Indirect cues: Body-like objects are forcibly opened, cut open or broken open.	“I also feel the scars, and my ribs hurt. Presumably because they had to break them for my surgery.”	1.00
Objects penetrate into the body/are devoured (incorporation)	Direct cues: The lung as object is in the body. Indirect cues: References to cannibalistic incorporation fantasies, any form of penetration, eating or swallowing.	“The new lung is now inside of me “; “It’s a bad thing because I need to swallow so many pills”; “It was as if masses of worms were coming at me and wanted to devour me.”	0.92
The donor is the recipient (identification).	Direct cues: Identification with the donor. Indirect cues: Any other form of equation or comparison with other persons or objects that indicates identification with the recipient.	“I feel like I am related to him”; “My old lung was exchanged for a new lung.”	0.92
The donor is a part of the recipient‘s internal world	Direct cues: Locating the donor in the patient’s inner world. Indirect cues: The donor is not located in the patient’s inner world, but e.g., in his organ; an object is contained in another object or vessel.	“It’s a part of me“; “It’s like there are two of me and I’m carrying someone else inside myself.”	0.96
The (new) object is a member of the family of organs	Direct cues: The lung is a member of the new family of organs. Indirect cues: Other statements about objects remaining in the body, other forms of not wanting to let go (e.g., constipation).	“I’ve adopted it lock, stock and barrel. It’s my lung now and I’m not giving it back”. “I have digestive problems and suffer from constipation”	0.86
The new organ (lung) can be expelled	Direct cues: Rejection of the lung, fear of rejection reactions. Indirect cues: Report of general excretion or loss of objects (e.g., diarrhea, vomit).	“I’m worried that a rejection reaction might occur”; “Yesterday I threw up, today I held it in”	0.86
Axis of Psychosomatic Totality			
Symbolic Pole	Physical symptoms as a compromise between desire and defense, referring to a latent imaginative content.	“My children understood my diagnosis to an extent. Suddenly, though, they began refusing their food and developed sleep problems. The pediatrician recommended talk therapy for my daughter.”	1.00
Asymbolic Pole	Physical symptoms are not organic; physical symptom has no symbolic function.		–
Organic Pole	Physical illness that may have resulted from stress, such as heart, circulatory and lung disease.	“And I’m breathing properly again, which I never really did before.”	1.00
Imaginary Zone	The imaginary space contains thoughts, images, ideas and feelings. In the imaginary zone, physical and affective experience is identical: Psychological pain is physical pain, psychological trauma is identical to physical injury.	“Yes, the old one could be thrown away. Yes, the new lung is also somewhere here in the center of me, it will always be like that.”	0.85

* Percentages are rounded to the second decimal place.

## Data Availability

The data presented in this study are available on request from the corresponding author. The data are not publicly available due to ethical guidelines.

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
