# Peer review of "The Actualization of the Transplantation Complex on the Axis of Psychosomatic Totality—Results of a Qualitative Study"

_healthcare, 2021, doi:10.3390/healthcare9040455_

Round 1
Reviewer 1 Report
The manuscript titled ‘The Actualization of the Transplantation Complex on the Axis of Psychosomatic Totality – Results of a Qualitative Study’ by Marie Eichenlaub et al., concerns very important psychological aspect of transplanatation medicine. The data analysis was focused on identifying unconscious and conscious feelings, including patients’ interviews and analysis of their postoperative dreams. It seems to be possible to take under consideration if some of the described unconscious processing could be related to the course of surgical intervention (for example, duration, additional complications, anesthesia). The paper reports that the development in psychotherapy is a valid challenge.
Author Response
To the Editors of Healthcare
Hamburg, March 31, 2021
Revised file (Manuscript ID: healthcare-1140812): The Actualization of the Transplantation Complex on the Axis of Psychosomatic Totality – Results of a Qualitative Study
Dear Editors,
Many thanks for your response and the valuable feedback from reviewers 1 and 2. Attached are the suggestions of the reviewers regarding form and content, and their implementation by the authors. The chapter and line numbers are given after each section in order to make the changes clear. The “Track Changes” mode in Word was also used.
An extensive revision of the language was suggested with aim of improving terminology and avoiding syntactic and grammatical errors. The authors have revised the text and passed it on to a qualified English-language translator for checking.
The suggestion of reviewer 2 to better operationalize what is meant by an “unstable state of physical health” was taken into account. An “unstable state of physical health” is taken to mean symptoms like dyspnea, inflammation, pain or fatigue that prevented the patient from taking part in the study. These additions can be found in Chapter 2.1 Sample and data (p. 3, lines 28-33).
Reviewer 2 also asked whether the patients were questioned as to their psychiatric history and possible psychopharmacological attitude before participating in the study, and the extent to which this might have led to skewed data. The authors also took this suggestion on board and describe how patients with severe psychiatric abnormalities such as dementia, psychosis or delirium were excluded before admission to the study (Chapter 2.1 Sample and data, p. 3, lines 33-35). The authors saw no reason to exclude patients suffering from other psychological stressors, since these patients also provide important information on how a lung transplant is mentally processed.
We are grateful for Reviewer 2’s valuable input via the question concerning the extent to which (pre-operative) psychopharmacological treatment or psychiatric disorders can affect the processing of the lung transplant. Since the answer would have involved an additional, relatively extensive analysis, we have refrained from examining this issue within the present paper. This limitation is now listed in Chapter 4 Discussion, Study Limitations and Strengths (p. 13, line number 45-48). The extent to which psychiatric disorders or preoperative psychopharmacological treatment might skew the results could be considered in a future analysis of the data. Here, we also referenced the study of Goetzmann and colleagues (2007), which shows that mental vulnerability may in fact be a predictor of a worse outcome (cf. Chapter 5 Conclusions, p. 14, line number 7-10).
In addition to the content-related comments, a typographical error (order of the citation in parentheses) was pointed out and taken into account. It was also suggested that we summarize the sociodemographic data in tabular form. The table can be found in Chapter 2.1 Sample and Data (p. 4, line number 2-43).
Besides the comments of the reviewers, it was pointed out in an email from Ms. Prdić (Assistant Editor) dated February 25, 2021 that the number of self-citations was too high. We therefore removed four publications that fell into this category from the References.
To our mind, the reviewers’ comments contributed to a significant improvement in quality of form and content of the paper The Actualization of the Transplantation Complex on the Axis of Psychosomatic Totality – Results of a Qualitative Study, for which we would like to express our sincere thanks.
We would be very pleased if the revised paper is now deemed suitable for publication in your journal.
Yours sincerely,
Marie Eichenlaub, Barbara Ruettner, Annina Seiler, Annette Boehler, Christian Benden, Uwe Wutzler and Lutz Goetzmann

Reviewer 2 Report
Dear Authors,
I found your study interesting. However, relevant methodological issues have been pointed out, which might affect the interpretations of the results.
Please, find attached the detailed comments.

Author Response

(The authors gave the same response as above.)

Round 2
Reviewer 2 Report
Dear Authors,
I appreciated your effort in providing a revised version of the manuscript. Moreover, you acknowledged the weakness of a limited psychiatric/psychological anamnesis. I encourage you to account for this relevant point in your future similar studies.
The manuscript currently offers an improved presentation of the study.
I endors its publication.